# Feasibility of a pragmatic randomized adaptive clinical trial to evaluate a brief negotiational interview for harmful and hazardous alcohol use in Moshi, Tanzania

**Catherine A. Staton**[1,2]*, **Kaitlyn Friedman**[2], **Ashley J. Phillips**[1], **Mary Catherine Minnig**[2], **Francis M. Sakita**[3], **Kennedy M. Ngowi**[4], **Brian Suffoletto**[5], **Jon Mark Hirshon**[6], **Monica Swahn**[7], **Blandina T. Mmbaga**[2,3,4,8], **Joao Ricardo Nickenig Vissoci**[1,2]

**1** Division of Emergency Medicine, Duke University School of Medicine, Durham, NC, United States of America, **2** Duke Global Health Institute, Duke University, Durham, NC, United States of America, **3** Kilimanjaro Christian Medical Centre, Moshi, Tanzania, **4** Kilimanjaro Clinical Research Institute, Moshi, Tanzania, **5** Department of Emergency Medicine, University of Pittsburgh School of Medicine, Pittsburgh, PA, United States of America, **6** Department of Emergency Medicine, University of Maryland School of Medicine, Baltimore MD, United States of America, **7** Department of Epidemiology, Georgia State University School of Public Health, Atlanta, GA, United States of America, **8** Kilimanjaro Christian Medical University College, Moshi, Tanzania

* catherine.staton@duke.edu

**Data Availability Statement:** Data cannot be shared publicly because of confidentiality and privacy concerns surrounding the human

## Abstract

### Introduction

Low-resourced settings often lack personnel and infrastructure for alcohol use disorder treatment. We culturally adapted a Brief Negotiational Interview (BNI) for Emergency Department injury patients, the "Punguza Pombe Kwa Afya Yako (PPKAY)" ("Reduce Alcohol For Your Health") in Tanzania. This study aimed to evaluate the feasibility of a pragmatic randomized adaptive controlled trial of the PPKAY intervention.

### Materials and methods

This feasibility trial piloted a single-blind, parallel, adaptive, and multi-stage, block-randomized controlled trial, which will subsequently be used to determine the most effective intervention, with or without text message booster, to reduce alcohol use among injury patients. We reported our feasibility pilot study using the *Reach*, *Effectiveness*, *Adoption*, *Implementation*, *Maintenance* (RE-AIM) framework, with recruitment and retention rates being our primary and secondary outcomes. We enrolled adult patients seeking care for an acute injury at the Kilimanjaro Christian Medical Center in Tanzania if they (1) exhibited an Alcohol Use Disorder Identification Test (AUDIT) ≥8, (2) disclosed alcohol use prior to injury, or (3) had a breathalyzer ≥0.0 on arrival. *Intervention* arms were usual care (UC), PPKAY, PPKAY with standard text booster, or a PPKAY with a personalized text booster.

participant data. Data are only available upon request as data transfer requires a written agreement approved by the National Institute for Medical Research (Tanzania). Data inquiries can be sent to Gwamaka W. Nselela at gwamakawilliam14@gmail.com.

**Funding:** This project was conducted with funding from the National Institutes of Health Fogarty International Center K01- TW010000-01A1 (PI Staton). Subsequently, the PI received further funding (1R01AA027512-01A1) from the Fogarty International Center to evaluate this intervention. The sponsor had no role in study design, data collection and analysis, decision to publish, or preparation of the manuscript.

**Competing interests:** The authors have declared that no competing interests exist.

## Results

Overall, 181 patients were screened and 75 enrolled with 80% 6-week, 82.7% 3-month and 84% 6-month follow-up rates showing appropriate *Reach* and retention. *Adoption* measures showed an overwhelmingly positive patient acceptance with 100% of patients perceiving a positive impact on their behavior. The *Implementation* and trial processes were performed with high rates of PPKAY fidelity (76%) and SMS delivery (74%). Intervention nurses believed *Maintenance* and sustainability of this 30-minute, low-cost intervention and adaptive clinical trial were feasible.

## Conclusions

Our intervention and trial design are feasible and acceptable, have evidence of good fidelity, and did not show problematic deviations in protocol. Results suggest support for undertaking a full trial to evaluate the effectiveness of the PPKAY, a nurse-driven BNI in a low-income country.

## Trial registration

**Trial registration number** NCT02828267. https://classic.clinicaltrials.gov/ct2/show/NCT02828267.

## Introduction

Alcohol use is associated with 3 million deaths annually [1,2] and is one of the leading avoidable risk factors for mortality and disability-adjusted life years (DALYs) lost globally. Alcohol consumption and the resulting disease burden is highest among low- and middle-income countries (LMICs) [3], where the consequences of harmful and hazardous alcohol use contribute to inequity between and within countries [1]. The WHO African Region has a 20% higher daily alcohol consumption among drinkers than the global average [1]. Tanzania reports a higher annual alcohol consumption per capita at 9.4L compared to the WHO African Region annual average of 6.4L [1]. Within Tanzania, studies have shown a high prevalence of hazardous alcohol use, particularly associated with lower socioeconomic status, unemployment, and male sex [4,5]. The prevalence of alcohol use is further higher in the Kilimanjaro region than that in other regions in Northern Tanzania, possibly due to differences in reporting behavior, socioeconomic status, or cultural and tribal practices [6,7].

Treatment of alcohol use disorders in LMICs, where it exists, has been focused on long-term residential treatment centers with limited overall efficacy and no evidence for moderate alcohol users with harmful or hazardous drinking [8–10]. Benegal et al. suggest an overall step-wise implementation of alcohol interventions, starting with opportunistic screening and brief advice and progressing to other, more complex treatment options [8]. Efficacious and cost-effective interventions are needed to reduce the alcohol-related burden in LMICs, and especially in low-income settings where these interventions are nearly absent [11]. A recent review found that 21 randomized, controlled trials have been conducted in 15 different LMICs between 1992 and 2018 with 86% of them employing motivational interviewing (MI) interventions, primarily in primary care settings [12]. Only two of these studies enrolled participants from an emergency department, both of which used MI principles but vastly different

interventions and did not compare MI to usual care. Segatto et al. compared a MI against an educational pamphlet control group [13]. While Sorsdahl and colleagues focused on a MI-Problem Solving combination treatment and thus was likely not powered to evaluate the MI only arm [14]. Overall, more research is needed in low-income settings, especially in EDs where acute complications of alcohol can be leveraged to motivate behavior change.

A recent Cochrane review demonstrated moderate evidence that brief interventions (BI) can reduce alcohol consumption in primary care or emergency care settings for harmful and hazardous drinkers with no adverse effects, while longer interventions had little additional effect [15]. A myriad of other high-income country (HIC) studies have shown that a BI administered specifically in an emergency department setting can reduce harmful alcohol use and its consequences up to 12 months post-injury [16–19] and decrease the incidence of trauma recidivism [20]. Brief negotiational interviews (BNIs) are a specific form of brief intervention that seek to help individuals who demonstrate risky alcohol use to identify and change their behavior by discussing personal motivations in a short interview [21,22]. While not intended for those with severe substance dependence, BNIs serve as a useful tool for moderating risky behaviors and can be administered by non-addiction specialized personnel [23–25].

BNIs could be incredibly useful treatments in low-resource settings, but the current optimal implementation strategy for the African injury population is unknown. BI studies in Africa have reported a range of effectiveness and are focused around specific subsets of the population [26,27]. There is also controversy around the best implementation strategy for a BI, such as the usefulness of boosters that serve as scheduled reminders post-BNI [28–30] and the integration of mobile health technology for LMICs [31,32]. Specifically, a Cochrane review found moderate evidence that digital interventions can reduce alcohol use by up to three standard units per week compared to control, which, given the low cost, can be quite beneficial in lower-resource settings [33]. Some studies have even explored the use of a personalized booster, created from information obtained in the original BI, in increasing the success of the intervention with limited additional costs [34,35].

"Punguza Pombe Kwa Afya Yako" (PPKAY)/ "Reduce Alcohol for Your Health" is a BNI created and piloted at the Kilimanjaro Christian Medical Centre in Moshi, Tanzania. PPKAY was based upon US-based BNI standards [36] and adapted to Swahili and the Tanzanian environment [37,38]. The effectiveness evaluation of this intervention will be with a future Pragmatic Randomized Adaptive Clinical Trial (PRACT) to determine if 1) BNI reduces alcohol-related harms at 3 months compared to usual care; 2) a standard SMS booster will improve the effectiveness of the PPKAY intervention; and 3) a personalized SMS booster is further effective than PPKAY with standard booster or PPKAY alone.

This study aims to assess the feasibility of our future PRACT at Kilimanjaro Christian Medical Centre in Moshi, Tanzania. Feasibility outcomes include recruitment rates, retention rates and acceptability of the intervention. The RE-AIM framework was used to report recruitment and retention rates, adoption and acceptability, uptake by clinical staff, implementation processes, and sustainability of the intervention [39].

## Materials and methods

### PRACT feasibility study design

PRACT is a three-stage single-blind parallel adaptive randomized controlled trial. Stage 1 compares usual care (UC) to PPKAY or PPKAY with standard booster through a 1:1:1 allocation with a 12-block randomization. Stage 2 removes the usual care arm but continues enrollment in the other two arms at a 1:1 allocation with a 12-block randomization. Stage 3 continues the most effective arm from Stage 2 and allocates additional participants into the most effective

arm and a new arm—PPKAY with a personalized booster. Additional participants are to be enrolled at a 1:4 allocation with 12-block randomization into the Stage 3 intervention arms. This adaptive method allows us to answer multiple successive questions with streamlined enrollment and processes; thus enrolling fewer patients in the fully powered trial than if we used three successive clinical trials. While for the future PRACT, all adaptations are planned to reach predetermined significance through interim analyses, the adaptations for this feasibility trial were triggered by enrollment numbers determined *a priori* (**Table 1**). Our processes of proceeding to the next stage mirrored our planned processes for the future PRACT. Similarly, as this is a feasibility study, stopping guidelines were not created and power considerations were not taken into account.

This feasibility study uses the RE-AIM framework to evaluate the *Reach*, *Effectiveness*, *Adoption*, *Implementation*, and *Maintenance* as delineated in **Table 2**. We conducted enrollment in a 75-person feasibility pilot to explore these objectives in order to prepare for our clinical trial.

## Study setting

This study was conducted at Kilimanjaro Christian Medical Center (KCMC). Located in Moshi, Tanzania, KCMC serves as the regional referral center for the Kilimanjaro Region, and the emergency department (ED) typically sees around 2,000 injury patients annually [40]. Of the adult injury patients who seek care in the KCMC ED, about 30% have a positive alcohol status upon arrival [40]. The KCMC ED reports a higher proportion of injury patients testing positive for alcohol compared to neighboring Mozambique (17%) [41], with alcohol use being associated with a 6-times increased risk of road traffic injury [42]. While rates of alcohol use are high, few testing and treatment resources exist [42]. In the KCMC community, mobile health technology is widely available, and a few other mobile health–based interventions have been successful focusing on vaccination [43], medication adherence [44,45], and counseling for different health services [46,47].

## Participants and eligibility criteria

Our pre-screening process identified adult patients (≥18 years of age) who were seeking initial care at the KCMC ED for an acute (<24 hours) injury regardless of gender. Of these patients, those who were not clinically intoxicated and who were capable of providing consent were

**Table 1. The RE-AIM framework with objectives.**

| RE-AIM Framework Elements | Feasibility Study Objectives |
|---|---|
| Reach | Patient eligibility rates<br>Patient recruitment rates<br>Patient retention rates |
| Effectiveness | Collect outcome measures including:<br>• Drinker Inventory of Consequences (DrInC)<br>• Alcohol Use Disorder Identification Test (AUDIT)<br>• Quantity and frequency of alcohol use (Q/F)<br>• Number of binge drinking days in the last month<br>Evaluate for potential negative effects |
| Adoption | Provider acceptability of the interventions<br>Patient acceptability of the interventions |
| Implementation | Interventions' fidelity<br>Trial process acceptability and adherence |
| Maintenance | Sustainability of the intervention<br>Resources to manage and implement the study and intervention |

**Table 2. Planned enrollment in feasibility of PRACT.**

| *Planned Feasibility PRACT Pilot Enrollment* | | | | | |
|---|---|---|---|---|---|
| | Usual Care (UC) | PPKAY | PPKAY with Standard Booster | PPKAY with Personalized Booster | Total |
| Stage 1 (1:1:1) | 10 | 10 | 10 | | 30 |
| Stage 2 (1:1) | | 10 | 10 | | 20 |
| Stage 3 (1:4) | | 0* | 4 ** | 16 | 20 |
| Total | 10 | 20 | 24 | 16 | 70 |

*Actual enrollment numbers may be different due to the nature of our adaptive sample size strategy, and

**continuation of the more effective arm.

offered informed consent. Patients who were intoxicated or too ill to provide consent were reassessed after 24 hours to determine capacity. Intoxication was determined clinically by the treating physician. During informed consent, three research nurses fully explained each study arm and the SMS text procedures to ensure that participants understood and were able to address any issues prior to providing consent. Written informed consent was obtained to test for alcohol consumption and for study participation. Patients were included in the study if they 1) disclosed alcohol use prior to injury, 2) scored ≥8 on the Alcohol Use Disorders Identification Test (AUDIT), or 3) tested positive (>0.00 g/dL) by breathalyzer. Participants remained eligible for recruitment even if they were found to have an AUDIT score >20, suggesting a severe alcohol use disorder (AUD). As there were no other treatment options, denying study participation to those with severe AUD was considered unethical. Participants were excluded if they did not speak Swahili, were too ill or otherwise unable to communicate, did not have an SMS-capable phone, or if they declined informed consent.

## Enrollment and retention processes

Our feasibility enrollment plan is delineated in **Table 1**. During the enrollment process, at least two phone numbers were identified and tested for follow-up contact. Nurses maintained contact with patients during the follow-up period by making phone calls to schedule follow-up assessments in advance. When needed, interviewers traveled to patients, provided financial compensation for travel expenses, or conducted surveys by phone.

## Randomization

Prior to study initiation, unblinded research personnel used computer software to randomize study identification numbers into 3 groups (UC, PPKAY, and PPKAY with Standard Booster) for Stage 1; 2 groups for Stage 2 (PPKAY or PPKAY with Standard Booster); and 2 groups for Stage 3 (the best performing arm in Stage 2 or PPKAY with Personalized Booster). We utilized block randomization for all three stages without any further restrictions. We chose a 12-block randomization to keep block size consistent through each stage, which allowed enrollment of even numbers of patients in each arm over the course of enrollment to reduce potential biases given the alcohol use differences per week, month, season or during holidays.

## Sample size

This feasibility study was created to pilot our adaptive trial methods and our study protocol; a sample size was determined based on feasibility of these methods not as a power analysis for effectiveness testing. Sample size for our feasibility enrollment was to mimic our trial

allocation on a smaller scale, and pilot the adaptation process between stages. We included at least 10 participants in each arm to test intervention/allocation protocols. Since this trial was aimed at assessing feasibility and was not powered to detect statistical differences in performance of each arm, we continued the PPKAY with the Standard Booster arm in Stage 3 according to hypothesized superiority.

### Allocation concealment and blinding

Allocation was concealed in order to avoid selection bias. Enrollment packets were placed in sealed opaque envelopes of equal size and thickness and locked in a drawer at the study site. Each envelope contained one paper denoting the random group assignment for each participant. Paper enrollment packets and data collection sheets were maintained to ensure reproducibility.

The future PRACT and this feasibility study are single-blind studies; our outcome assessors were blinded to participant allocation. We have established a protocol where one research team member performs initial and final outcome assessments while a nurse obtains the randomization packet and performs the corresponding intervention.

### Interventions

**PPKAY.** PPKAY is a nurse-administered, 5–30 minutes BNI based on "SBIRT Educational Toolkit"' using the FRAMES motivational interviewing techniques [48,49]. This is a four-step discussion: 1) raise the subject of alcohol, 2) provide feedback, 3) enhance motivation, 4) negotiate and advise. PPKAY was culturally adapted based on quantitative and qualitative data from the Swahili-speaking KCMC injury population [37]. Prior to feasibility trial enrollment, PPKAY was iteratively improved and reassessed by Tanzanian nurse counselors who evaluated the cultural appropriateness and the adherence to the tenants of BNI and motivational interviewing.

The PI, or a delegate trained by the PI, reviewed PPKAY training protocol documents on the FRAMES motivational interviewing technique to train nurses. Mock interviews were completed after training to prepare the nurses to deliver this tool appropriately. Nurses conducted pre-intervention screening with our locally validated AUDIT tool, alcohol use questions, and breathalyzer testing [50]. They approached patients and asked permission to raise the subject of alcohol and discuss their current alcohol use compared with local and international safe use guidelines. Next, nurses assessed the patient's readiness to change with a (0–10) readiness-to-change ruler, adapted to Swahili and the local context through iterative research based on nurse content evaluation and comprehension assessments. They then identified a patient's specific reasons for changing behavior (social, family, financial, etc.) and compared it with their Readiness Ruler response, highlighting self-efficacy and personal capacity for behavior change. Finally, they negotiated a reduction in alcohol use, optimally lower than safe use guidelines. At the close of the PPKAY, the provider thanked the patient for their openness in discussing alcohol, promoting self-efficacy, and fostering a positive patient–provider relationship.

**PPKAY with standard booster.** Patients allocated to PPKAY with Standard Booster underwent the same PPKAY process listed above. Patients provided a phone number where they could be reached, and, prior to patient discharge, this cell phone was tested by the study team to ensure it was working. After hospital discharge, one of the four standard motivational text messages was sent to participants' cell phones (rotating messages weekly) until conducting the final follow-up.

**PPKAY with personalized booster.** Patients allocated to the PPKAY with Personalized Booster underwent the same PPKAY and text booster processes listed above. After completing

the PPKAY, nurses used the PPKAY content, specifically the person's unique reasons for change, to complete *a priori* established outlined texts. These texts were entered into our text message system by the researchers on the day of PPKAY and were kept for evaluation by the research team to review at the weekly quality control meetings. After hospital discharge, one of the four personalized motivational text messages was sent to participants' cell phones (rotating messages weekly) until the final follow-up was conducted.

**Usual care (control).**   Our pragmatic goals included understanding the benefit of PPKAY compared to the current standard practice or usual care (UC). In the UC study arm, patients did not receive any discussion or instructions on alcohol or alcohol reduction, feedback on their AUDIT scores, or the PPKAY intervention. Patients only received normal discharge instructions pertaining to their injury and were not contacted by research staff, except for follow-up visits.

All patients in our feasibility trial are followed up on the same follow-up schedule at 6 weeks, 3 months, and 6 months. According to prior literature on return to alcohol use, we expected to see an impact from our intervention at 3 months [50–54].

## Feasibility evaluation outcomes

**Reach (Patient eligibility, recruitment and retention rates).**   We recorded the total number and rate of patients pre-screened, the proportion who provided consent and reasons for not consenting, which eligibility criteria were met, the total number randomized, and the reasons for ineligibility or exclusion. Based on prior rates of alcohol use in this population [42], our benchmark was 30% of pre-screened patients to be randomized with an average of 5 patients per week. Retention rates were calculated as the proportion of randomized participants who completed each assessment time-point (baseline, 6 weeks, 3 months, and 6 months) with a 60% retention rate at each time as a benchmark for feasibility.

**Effectiveness.**   To demonstrate a potential positive effect (benefit) and limited negative effect (risk), we collected outcome measures corresponding to the future PRACT. The primary outcome measure was the number of binge drinking days with secondary outcomes being Drinker Inventory of Consequences (DrInC) score [55,56], quantity and frequency of alcohol use, and AUDIT score [50,57]. All quantitative data were reported in standard drinks, where one standard drink is equal to roughly 14 grams of pure alcohol. The research team is extensively trained in the collection of alcohol use data, including the number and size of alcohol containers, and the types of alcohol to determine the approximate grams of absolute alcohol or 'standard drink' [42]. For 'home brew', alcohol content can be incredibly variable from 2% to 20% alcohol by volume and increases with shelf-life. As such, we estimated 5% alcohol by volume and container sizes that are typical for the region [58]. We were not able to pursue objective (e.g., assay testing) outcome verification given availability, cost, and feasibility limitations. Potential negative effects included increased alcohol use or potential loss of confidentiality associated with the intervention. However, it is important to note the aim of the current study was to assess the feasibility of the intervention and not the effectiveness with respect to clinical outcomes.

**Adoption (patient & provider acceptability).**   Acceptability of the intervention was assessed from participants in the intervention arms (PPKAY, PPKAY with standard and personalized boosters) with 15 questions in a post-treatment survey and interview. We asked 5 questions about satisfaction with various domains as well as perceived usefulness, each with structured response options. We also asked 4 open-ended questions on the overall structure and content of the intervention.

During post-trial debriefing, we conducted a focus group discussion with our research nurses and research team. During this discussion, after informed consent, nurses were asked about the complexity of the intervention, eagerness to administer PPKAY, and workload. The focus group sessions were audio recorded and transcribed. We used inductive coding to understand the nurses' perceptions about the overall trial protocol, patient and nurse acceptability of the intervention, trial process feasibility and fidelity, and perceived impact of the intervention.

**Implementation.** *Intervention Fidelity*. Other pragmatic trials on alcohol harm reduction strategies, especially in Emergency Departments, have exhibited limited intervention fidelity [59]. As such, we incorporated an intervention fidelity evaluation and quality improvement (QI) process. In prior work, our team adapted and validated the BI Assessment Scale [60] to the Tanzanian setting. It has good psychometric properties including inter- and intra-rater reliability with Kappa and ICC values above 0.80 and a strong ability to differentiate standard PPKAY sessions with poorly performed PPKAY sessions. During our study, we audio-recorded the interventions so that they could be evaluated weekly by bilingual researchers to determine adherence to the intervention plan using our validated scale. For QI, a Swahili-speaking researcher listened to recordings of each intervention, graded it on the BI Assessment Scale and provided nurses with feedback to improve future interventions. Regular debriefing meetings were held with the study team to discuss the process of intervention implementation. Similarly, our SMS booster intervention arms were evaluated to determine intervention fidelity; we evaluated whether text messages were sent and why some messages were unable to be delivered. We obtained some patient responses to our text boosters which were translated and a thematic analysis was performed.

*PRACT Processes*. We assessed whether the research team could effectively randomize participants into trial arms, maintain blinding, and whether participants accepted their assigned condition. We assessed this process by the proportion of assigned intervention sessions attended, with a benchmark of 80%. We also assessed our feasibility PRACT process by conducting an evaluation of protocol deviations.

**Maintenance (sustainability).** We conducted a focus group with providers to assess the acceptability to conduct the interventions and determine the perceived time burden, as a measure of sustainability. Furthermore, we evaluated the resources available for the research team to manage and implement the PRACT's innovative trial design.

## Statistical analysis considerations

This study attempted to determine the feasibility of conducting adaptations in this setting; however, effect size-based stopping rules were not applied, as this feasibility lacked the power to reach adaptation parameters. Adaptation feasibility was determined through the number of enrolled participants. Our study was not designed to determine the effectiveness of the intervention and therefore the sample size was not powered to determine an effect size. Outcomes were collected at baseline and throughout the follow-up periods to demonstrate the feasibility of collection at these time points.

Given the nature of this study, we focused on descriptively reporting the results of our feasibility and outcome assessments. We report numeric data as means and standard deviations, while categorical data are reported as absolute and relative frequencies. Baseline patient characteristics are described for all arms using descriptive statistics. Outcome data are reported for baseline, 6 week, 3 month, and 6 month follow-up for each intervention arm.

### Ethical statement

This study was approved by the Duke University Medical Center Institutional Review Board (Pro00062061), Kilimanjaro Christian Medical Center Ethics Committee and the Tanzanian National Institute of Medical Research. The pilot study is registered as *"Evaluating a Brief Negotiational Intervention for Alcohol Use Among Injury Patients in Tanzania (BNI)"* with ClinicalTrials.gov registration number is NCT02828267.

### Changes to protocol from project inception

The initial proposal for the current study evolved from Dr. Staton's initial K01 award application to its final form on the basis of years of preliminary research [7,37,42,61–63]. Originally envisioned as a single-stage, two-armed feasibility trial, this feasibility pilot study ultimately featured an adaptive three stages plan and four arms. Of note, booster messages and alcohol-related outcomes were not included in the original protocol. Changes to the protocol occurring during contextual and cultural input during intervention creation have been published on ClinicalTrials.gov and approved by all relevant ethics bodies.

### Inclusivity in global research

Additional information regarding the ethical, cultural, and scientific considerations specific to inclusivity in global research is included in the (S1 Checklist).

## Results

### Reach

**Patient eligibility, recruitment and retention rates.** Seventy-five patients were enrolled over the course of 12 weeks, between July 3, 2018, and September 21, 2018 (average 6.25 patients per week) (**Fig 1**). All patients who presented to the ED were pre-screened, 181 of whom consented for pre-randomization; of these, 41% (n = 75) met inclusion criteria and were randomized. Three participants were discharged prior to administration of the intervention, but we included those enrolled in an intention-to-treat and pragmatic approach to enrollment. Most participants were men (96.0%) who were married (68.0%), with mean age 36.3 years (SD 13.5) and of the Chagga tribe (60.0%) (**Table 3**). Retention rates were maintained at a minimum of 70% across each intervention arm at 6-week, 3-month, and 6-month follow-up times for a final 6 month retention rate of 84% across all arms.

### Effectiveness

As seen in **Table 4**, there was a decrease in the DrInC and the AUDIT score across all groups from baseline to 6 weeks, 3 months, and 6 months. Although a pilot, at 6-month, participants in the intervention groups showed a higher decrease in average DRINC and AUDIT scores than the usual care group. We also assessed for any potential negative effects of the interventions, including whether increased drinking was observed in any participants or whether confidentiality was lost at any point in the study, and none were observed.

### Adoption

**Patient acceptability.** At 6-month follow-up when asked specifically about the PPKAY or the SMS boosters, 100% (n = 54) of respondents surveyed believed that the PPKAY had a positive impact on their drinking behavior, with 98% reporting a *large* positive impact. Similarly, 100% of participants believed that the SMS boosters had a positive impact on their behavior,

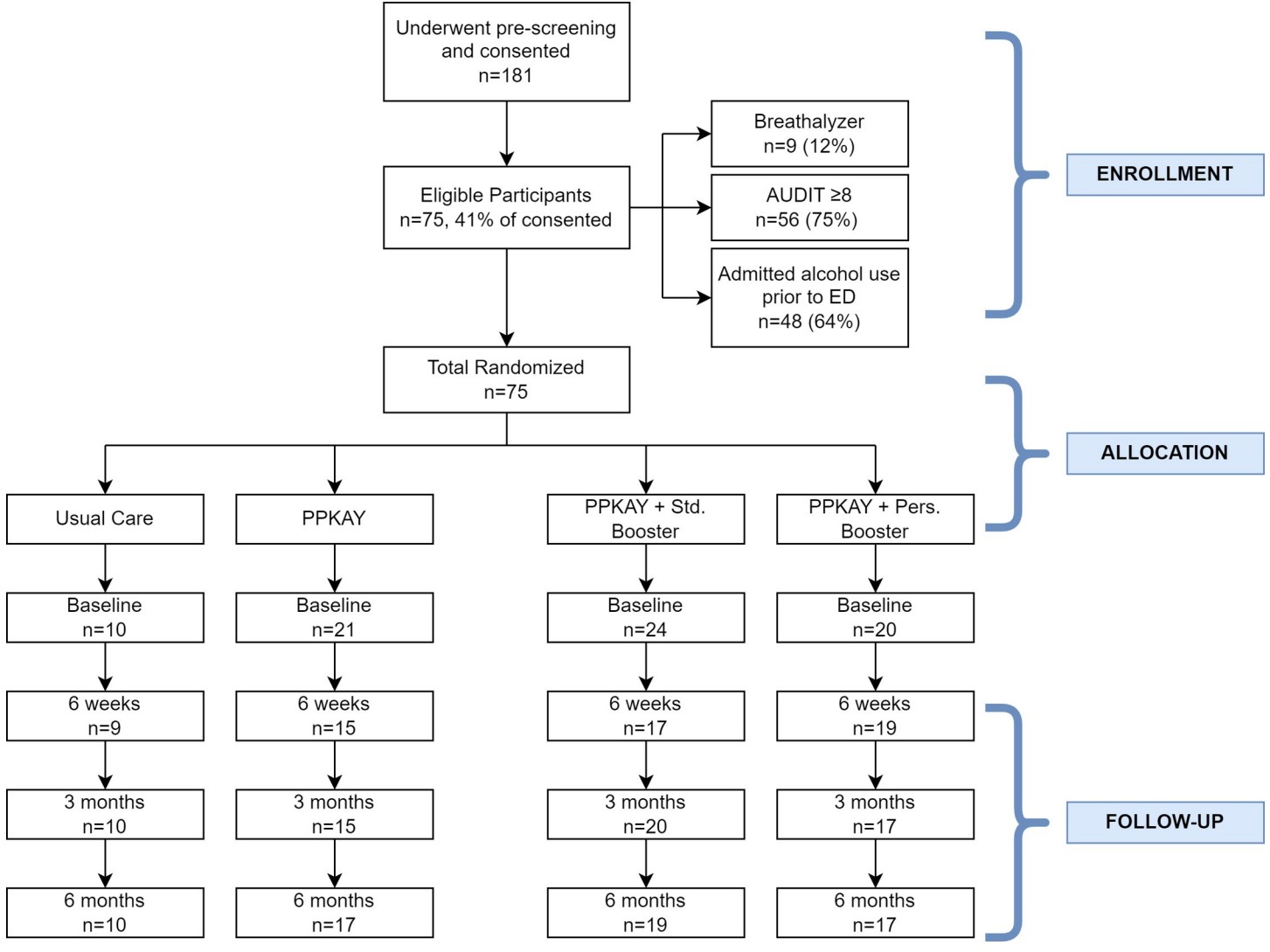

**Fig 1. Eligibility, recruitment, and retention flowchart.**

with 91% reporting a *large* positive impact. Eighty-two percent of participants in the SMS intervention arms reported that they were pleased with the timing of when the text messages were sent out because they had time to read the messages right away. Eighty-six percent of respondents believed that the BNI took place at a convenient time for them, with the remainder of participants reporting that they were in too much pain to complete the intervention at designated time. All participants reported that they believed nurses were the correct care providers to administer this type of intervention (see **S1 Table**).

**Provider acceptability.** Nurses stated they were eager to administer PPKAY. Nurses believed that the majority of patients enthusiastically participated in the PPKAY, indicating that they believed a full future PRACT trial would be successful since the participants "really like[d] . . . the counselling process" and even indicated a desire that the patient's spouse receive counselling as well. Nurses also made it clear that they felt confident about their ability to conduct this study in a full future trial, and that they "will have time" to add the PPKAY to their workload.

**Table 3. Demographics of those eligible for follow up.**

| | Usual Care n = 10 | PPKAY n = 21 | PPKAY with Standard Booster n = 24 | PPKAY with Personalized Booster n = 20 | All Groups N = 75 |
|---|---|---|---|---|---|
| Age, years M (SD) | 36.7 (12.8) | 40.6 (13.4) | 36.6 (14.5) | 31.4 (11.8) | 36.3 (13.5) |
| Male (n,%) | 10 (100%) | 19 (90.5%) | 23 (95.8%) | 20 (100%) | 72 (96.0%) |
| Years of education mean (SD) | 9.7 (3.6) | 9.6 (3.1) | 8.4 (2.6) | 8.7 (2.5) | 9.0 (2.9) |
| **Tribe,* n (%)** | | | | | |
| Chagga | 6 (60.0%) | 11 (52.4%) | 16 (66.7%) | 12 (60.0%) | 45 (60.0%) |
| Saamba/ Maasai | 0 | 0 | 2 (8.3%) | 0 | 2 (2.7%) |
| Pare | 2 (20.0%) | 5 (23.8%) | 1 (4.2%) | 4 (20%) | 12 (16.0%) |
| Other | 2 (20.0%) | 5 (23.8%) | 5 (20.8%) | 4 (20%) | 16 (21.3%) |
| **Marital Status, n (%)** | | | | | |
| Single | 2 (20.0%) | 5 (23.8%) | 7 (29.2%) | 6 (30.0%) | 20 (26.7%) |
| Married | 8 (80.0%) | 15 (71.4%) | 15 (62.5%) | 13 (65.0%) | 51 (68.0%) |
| Widowed/ Separated | 0 | 1 (4.8%) | 2 (8.3%) | 1 (5.0%) | 4 (5.3%) |
| **Employment, n (%)** | | | | | |
| Professional | 1 (10.0%) | 3 (14%.3) | 3 (12.5%) | 3 (15.0%) | 10 (13.3%) |
| Skilled Employment | 3 (30.0%) | 2 (9.5%) | 5 (20.8%) | 6 (30.0%) | 16 (21.3%) |
| Self-employed | 4 (40.0%) | 8 (38.1%) | 5 (20.8%) | 9 (45.0%) | 26 (34.7%) |
| Farmer | 2 (20.0%) | 8 (38.1%) | 8 (33.4%) | 2 (10.0%) | 20 (26.7%) |
| Student/Other | 0 | 0 | 3 (12.5%) | 0 | 3 (4.0%) |
| **Monthly Income** | | | | | |
| TSH, mean (SD) USD, mean (SD) | 201,000 (101,811) 89 (45) | 187,947 (147,249) 83 (64) | 178,863 (144,896) 79 (64) | 197,315 (111,386) 87 (49) | 185,500 (129,257) 83 (57) |
| **Alcohol Use Information, n (%)** | | | | | |
| BAC positive on arrival | 1 (10.0%) | 4 (19.0%) | 3 (12.5%) | 1 (5.0%) | 9 (12.0%) |
| AUDIT ≥ 8 | 8 (80.0%) | 18 (85.7%) | 19 (79.2%) | 14 (70.0%) | 59 (78.7%) |
| Self-reported alcohol use prior to injury | 5 (50.0%) | 16 (76.2%) | 14 (58.3%) | 15 (75.0%) | 50 (66.7%) |

*Proportions may not total 100% due to rounding.

**1 USD = 2271 TSH.

**Table 4. Comparison of outcomes in the week prior to follow-up.**

| Scale | Follow-up period | Usual Care | PPKAY | PPKAY with Standard Booster | PPKAY with Personalized Booster |
|---|---|---|---|---|---|
| DrInC Score mean (SD) | Baseline* | 14.7 (8.0) | 20.3 (7.3) | 16.0 (10.3) | 15.0 (8.7) |
| | 6-wk Follow-up** | 0.2 (0.7) | 0.9 (1.8) | 1.9 (7.8) | 0.3 (0.9) |
| | 3-mo Follow-up** | 0.2 (0.6) | 0.4 (1.1) | 2.4 (8.5) | 0.7 (1.2) |
| | 6-mo Follow-up** | 1.3 (3.5) | 1.2 (3.2) | 1.0 (3.1) | 0.7 (1.5) |
| AUDIT score mean (SD) | Baseline* | 9.7 (3.8) | 13.9 (6.4) | 12.1 (6.5) | 12.9 (7.2) |
| | 6-wk Follow-up** | 0.33 (1.0) | 1.5 (2.5) | 1.9 (5.3) | 1.2 (1.4) |
| | 3-mo Follow-up** | 0.7 (0.9) | 1.9 (1.3) | 2.9 (5.0) | 2.7 (1.4) |
| | 6-mo Follow-up** | 4.4 (5.3) | 4.0 (4.1) | 3.3 (2.3) | 4.9 (2.6) |

*n = 75, follow up in the last year

**since follow up.

## Implementation

**PPKAY feasibility and fidelity.**   Nurses found the PPKAY simple and easy to administer, allowing a maintained high rate of intervention fidelity. They believed patients were interested in receiving this information and they were eager to participate in the process. Similarly, nurses stated the intervention was not burdensome. Nurses maintained a high rate of PPKAY intervention fidelity. Overall, 76% of the PPKAY interventions delivered had 19 or more of the 22 critical actions as recommended in the BI Assessment Scale [37,60]. Interventions performed after QI discussions were found to be more adherent than earlier ones, suggesting success of our QI processes.

**SMS feasibility.**   In total, 44 patients were supposed to receive 25 messages over 6 months, but due to a staff communication error, texts were only sent for 3 months or 12 messages or 528 total texts. We had 3 participants who left prior to their intervention and therefore were not enrolled in the SMS text system; therefore, we would expect 492 texts attempted. Our system demonstrated a total of 258 texts (52% of expected) were attempted by the end of the 3-month follow-up period. This difference highlighted the need for more close weekly monitoring of our SMS system. No further understanding of this process was found during the feasibility trial but informed our processes for the full trial. Seventy-four percent of attempted texts were registered as sent through the messaging system. This send rate was found to be equal across each intervention arm. Of the text messages that failed delivery, most were not delivered because the participant traveled outside of cellular coverage or had their phone off, prohibiting delivery. Our only protocol deviation was termination of SMS text messages at 3 months post-intervention rather than 6 months post-intervention due to limited communication.

**Feasibility PRACT process evaluation.**   There were two errors in the envelope randomization processes, where patients were allocated to an intervention arm other than their planned arm. Throughout the entire study duration, only 2 patients who were assigned PPKAY did not receive the intervention, as they were discharged prior to receiving the intervention. These two patients were followed as normal as per protocol of the planned intention to treatment analysis. There was no reported loss of confidentiality and no increase in any alcohol-related measures. In two instances, the audio tape recordings failed to record patient interview sessions to assess intervention fidelity.

## Maintenance

Providers reported that screening for eligibility and providing the interventions lasted an estimated 30 minutes per patient. There was agreement between providers and nurses that this time commitment would be feasible for trained nurses to continue in the future.

## Discussion

To our knowledge, this is the first feasibility study of an adaptive clinical trial evaluating a culturally adapted brief negotiational interview for injury patients in a low income setting. While BNI has been studied and shown to successfully reduce alcohol-related harms in numerous HIC studies [64] and in LMIC [65], this is the first study to examine the feasibility of conducting such a trial in a low-income country—Tanzania. We used the RE-AIM framework as a way to synthesize our results, which demonstrated successful trial feasibility, including good recruitment and retention, high intervention fidelity, and high rates of patient and provider acceptability. Overall, these findings indicate that a full PRACT trial could be successfully conducted in the future. Similarly, we found a positive trend of the PPKAY being effective at reducing alcohol-related harms.

## Trial feasibility: Reach, trial processes

**Reach.** Our study population primarily consisted of healthy men. This is because injuries and alcohol use are primarily observed among men in this setting. The study demographic is expected based on prior studies [40,42]. While young healthy men seen in an ED are traditionally a difficult population for follow-up, our enrollment rates were relatively high (41%, 75 of 181 screened) compared to Sorsdahl's ED-based brief intervention clinical trials (12%, 335 of 2736) in South Africa [14]. Similarly, our 6-month follow-up rate at 84% is promising when compared to Sorsdahl's South African trial, where 54% completed 3-month follow-up [14], or Segatto's 85.2% 3-month follow-up in Brazil [13]. Activities described in the methods section may have contributed to high rates of follow-up. Additionally, SMS messages may have reminded assigned individuals of their ongoing participation in the trial, perhaps contributing to their willingness to continue participation.

**Trial processes.** A traditional approach to investigating an intervention implementation package would include multiple, successive randomized clinical trials, each with specific research questions requiring significant time and financial resources [66]. Our project aims to use an adaptive trial to show effectiveness over usual care while ensuring that most patients can receive this potentially beneficial intervention. Similarly, our innovative PRACT highlights the strengths of an adaptive trial using an adaptive method during the preliminary phases where we expect a large effect size, but not during the final phase where we expect to find smaller effect sizes. While adaptive clinical trials are gaining popularity, they have rarely been used in behavioral science or global health. It is in these arenas where these methods would be most advantageous to expedite the process of discovery. Given our goal to use adaptive methods, conducting a feasibility trial to pilot our methods and processes was critical to the trial's success. Our analyses found that improving communication for our multi-national team and organization and pre-planning for our blinding and randomization processes are paramount; but it also showed these processes are feasible in our setting, as we had only minimal protocol deviations.

## Intervention feasibility: Maintenance, efficacy, intervention

**Maintenance.** Challenges to intervention feasibility identified in HIC settings have mainly included sustainability concepts. Concerns surround limited nursing staff time to conduct the intervention, especially in a busy ED, and perception of preparedness to discuss harm reduction strategies with participants [64]. Interviews conducted with nursing staff in our feasibility trial indicated that BNI would not add excessive burden to their commitments. Our nurses also believed the training they received was sufficient, and that other counselors would be ready to conduct the intervention with the same training. The PPKAY intervention was highly accepted by both nurses and patients. Nurses reported that they were eager to administer the intervention and indicated that patients received the PPKAY positively. Patients reported benefits from both the standard booster and personalized booster. Both nurses and patients responded positively to PPKAY, potentially due to familiarity. After efficacy testing, widespread dissemination of this protocol should consider champions on the ground to foster strong familiarity and continue positivity.

**Efficacy.** This study found that the PPKAY had a self-reported positive impact on the behavior of all participants. While this trial was not powered to determine an effect size, we were able to see a signal of change among the intervention groups. Hence, we are optimistic of the impact of this intervention, especially given the overwhelmingly positive feedback we received about the patient acceptability of the intervention and preliminary findings. Given

our preliminary efficacy findings, there is a strong potential in a longer-term intervention to reduce alcohol-related harms in the Tanzanian injury population.

Previously, studies examining the feasibility and efficacy of SMS message boosters in BIs for alcohol use have taken place in HICs [33,67], but evidence is lacking on their use in LMICs, such as Tanzania. This study showed high intervention fidelity with the added ability of mobile health technology integration. With high rates of deliverable SMS messages, nurse acceptability, and successful quality improvement actions, a pragmatic trial using SMS messages appears feasible. Our only protocol deviation occurred with a communication error where SMS were sent for 3 months after enrollment rather than 6 months post-enrollment. Otherwise, each step of our pilot intervention occurred smoothly, without placing extra burdens on the research and nursing staff conducting the study. Our innovative and previously untested SMS booster functioned well, required low maintenance, and was implemented easily. Since we experienced so few difficulties implementing the BNI and SMS interventions, we believe a pragmatic trial will also be able to be performed successfully.

## Limitations

This study serves as an initial feasibility trial, and thus carries limitations within the study design in terms of size, scope, and generalizability. Given that this was a preliminary study, the sample size was not powered to determine an effect size for outcomes. Thus, the clinical significance of results must be cautiously interpreted. In terms of generalizability, it is important to consider the demographic and patient profile of this study, and that the results are likely not relevant to higher-income country settings. It is possible that our participant population does not fully reach individuals in the lowest income categories since participants included in this study must have mobile phone access, and the financial resources and ability to reach ED. That said, this is a pragmatic trial at a regional trauma center where free referrals from other public clinics and free care can be provided, and is intended to serve as a proof of concept pragmatic effectiveness trial. We found exceptionally high rates of patient and provider acceptability of the PPKAY and SMS messages. It is important to note, however, that response bias is possible given that interviews were conducted by the study PI, the research nursing staff's employer. In an attempt to address this potential bias, nursing staff were asked directly about the specific barriers they faced during enrollment, randomization, and follow-up stages.

## Conclusion

This feasibility trial demonstrated successful recruitment, enrollment, and follow-up in a pilot pragmatic randomized adaptive clinical trial (PRACT) of a nurse-led BNI aimed at reducing harmful alcohol use in a low-income country emergency department. Integration of SMS mobile health technology was also shown to be feasible and acceptable in this population. These promising results recommend a full-scale trial to determine the effectiveness of PPKAY BNI with SMS booster in reducing harmful alcohol use.

## Supporting information

**S1 Checklist.**
(DOCX)

**S2 Checklist.**
(DOCX)

**S1 Table. Results of patient acceptability qualitative interviews.**
(DOCX)

## Acknowledgments

We would like to acknowledge our KCMC Emergency Department Research team who have stewarded our research over the past 10 years. Their unwavering dedication to their patients and community are what drive us to continue to work with and for the Moshi community.

**Ethics**

The current study was approved by the Duke University Health System Institutional Review Board (Pro00062061), the Kilimanjaro Christian Medical University College Research Ethics Review Committee (Clearance Certificate #561), and the Tanzania National Institute for Medical Research National Health Research Ethics Committee (Ref. NlMRIHQIR.8aNol.IXl2121). Written informed consent was obtained from all participants.

## Author Contributions

**Conceptualization:** Catherine A Staton, Brian Suffoletto, Jon Mark Hirshon, Monica Swahn, Blandina T Mmbaga, Joao Ricardo Nickenig Vissoci.

**Data curation:** Ashley J Phillips, Mary Catherine Minnig, Kennedy M. Ngowi.

**Formal analysis:** Ashley J Phillips, Mary Catherine Minnig, Joao Ricardo Nickenig Vissoci.

**Funding acquisition:** Catherine A Staton.

**Investigation:** Kennedy M. Ngowi.

**Methodology:** Joao Ricardo Nickenig Vissoci.

**Project administration:** Catherine A Staton, Ashley J Phillips.

**Resources:** Kennedy M. Ngowi.

**Software:** Kennedy M. Ngowi, Joao Ricardo Nickenig Vissoci.

**Supervision:** Catherine A Staton, Francis M Sakita, Blandina T Mmbaga.

**Writing – original draft:** Catherine A Staton, Kaitlyn Friedman, Ashley J Phillips, Mary Catherine Minnig.

**Writing – review & editing:** Catherine A Staton, Kaitlyn Friedman, Francis M Sakita, Kennedy M. Ngowi, Brian Suffoletto, Jon Mark Hirshon, Monica Swahn, Blandina T Mmbaga, Joao Ricardo Nickenig Vissoci.

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
