## [Decision Letter · Decision Letter 0]

10 Oct 2022

PONE-D-21-39881Feasibility of a Pragmatic Randomized Adaptive Clinical Trial to Evaluate a Brief Negotiational Interview for Harmful and Hazardous Alcohol Use in Moshi, TanzaniaPLOS ONE

Dear Dr. Staton,

Thank you for submitting your manuscript to PLOS ONE, and for revising the manuscript in response to the reviews for the previous submission of this work. After careful consideration, we feel that it has merit but does not fully meet PLOS ONE’s publication criteria as it currently stands. Therefore, we invite you to submit a revised version of the manuscript that addresses the points raised during the review process. This submission has been evaluated by three reviewers whose comments are included below. Please address all of the reviewers' comments (except Reviewer 1, point 1) through revisions to your manuscript and in a Response to Reviewers document. Reviewer 1's first comment raised, "Authors have mentioned that the aim of the study is to assess the feasibility however the title of study mentions it as randomized controlled trial." As your title clearly identifies the study as a feasibility study, you can disregard this comment. You addressed a sample size calculation query in your Response to Reviewers section, and this issue was again raised for the current submission, by Reviewer 2. From your Response to Reviewers comments it is my understanding that you did not do a formal sample size calculation for this study. In the Sample Size section of the Methods, please clarify the approach taken in deciding upon the sample size (i.e. the rationale for the group sizes used), and transparently state whether a formal sample size calculation was not performed. Also, please discuss this issue in the Limitations section in the context of the references cited by Reviewer 2 for the prior submission and Reviewer 2 for the current submission. In the Response to Reviewers you explained that there were differences between the original approved study protocol, manuscript, and Clinical Trials registry for this study. Please add a section to the Methods section that discusses any changes from the original approved protocol; explains any discrepancies between the manuscript, original protocol, and registry; and notes if/when any changes were approved by the IRB.  Please submit your revised manuscript by Nov 22 2022 11:59PM. If you will need more time than this to complete your revisions, please reply to this message or contact the journal office at plosone@plos.org. Please include the following items when submitting your revised manuscript:A rebuttal letter that responds to each point raised by the academic editor and reviewer(s). You should upload this letter as a separate file labeled 'Response to Reviewers'.A marked-up copy of your manuscript that highlights changes made to the original version. You should upload this as a separate file labeled 'Revised Manuscript with Track Changes'.An unmarked version of your revised paper without tracked changes. You should upload this as a separate file labeled 'Manuscript'.

We look forward to receiving your revised manuscript.

Kind regards,

Renee Hoch, Ph.D.

Managing Editor, PLOS Publication Ethics

PLOS ONE

Journal Requirements:

Reviewers' comments:

Reviewer's Responses to Questions

**Comments to the Author**

1. Is the manuscript technically sound, and do the data support the conclusions?

Reviewer #1: Yes

Reviewer #2: Partly

Reviewer #3: Yes

2. Has the statistical analysis been performed appropriately and rigorously? 

Reviewer #1: No

Reviewer #2: No

Reviewer #3: I Don't Know

3. Have the authors made all data underlying the findings in their manuscript fully available?

Reviewer #1: Yes

Reviewer #2: No

Reviewer #3: No

4. Is the manuscript presented in an intelligible fashion and written in standard English?

Reviewer #1: No

Reviewer #2: Yes

Reviewer #3: Yes

5. Review Comments to the Author

Reviewer #1: Dear Author,

The manuscript has covered feasibility of important intervention in low resource setting. There are some major and minor comments which need to be addressed before it can be considered for the publication.

1. Major comments: Authors have mentioned that the aim of the study is to assess the feasibility however the title of study mentions it as randomized controlled trial which is inherently supposed to measure effectiveness of the intervention vis a vis control group. So authors must do statistical analysis to see if there is any significant or non significant differences in outcome measures using non-parametric tests since the sample size may not be enough for parametric tests. Otherwise it would be wrong to refer the present manuscript as RCT. The paper can be best described as feasibility study and not RCT. So please update results sections to compare outcome measures accordingly. The findings should also be discussed later.

2. Results section does not mention about findings of 15 questions in a post-

treatment survey/interview and 5 questions about satisfaction

3. The trial has higher retention rate, the reason for higher retention rate should be highlighted (e.g. home visits by investigators, financial incentives given to the project staff and number of times follow up calls were attempted. All these issues are important when intervention is used in the clinical setting

4. The authors have not mentioned what treatment was offered for patients with alcohol use disorders, any psychiatrist consultation was sent ?, referred to outpatient department for management of the problem?

5. The timing of intervention is ranged between 5 minutes to 30 minutes. It is important to mention average time of intervention. As this raises question about fidelity of intervention. It will be goo to mention number of nursing professionals who were involved in delivery of intervention, any family members involved as some patient may be brought by law enforcement agencies while some may brought by family members. Involvement of family members are likely to impact outcome measures.

6. Discussion section need to be expanded to discuss the findings in reference to available literature from the region. Also clinical implications of the study if any.

Regards,

Reviewer #2: The mansucript explores feasibility of a pragmatic randomized adaptive controlled trial of the PPKAY intervention among subjects recruited in Tanzania. The study objectives are clearly stated; the authors proposed a block-randomized controlled trial. In future, the effectiveness of the intervention will be assessed via a pragmatic randomized adaptive trial. My comments are as follows:

1. I still think a sample size/power statement would make the approach more rigorous. Just stating that someone included 10 participants in each arm do not make a strong case. Samples sizes for pilot/feasibility studies are requested; see below:

https://www.rds-london.nihr.ac.uk/resources/justify-sample-size-for-a-feasibility-study/

Also, the 3-stage design as proposed is not so trivial.

Reviewer #3: 1. The data appear to support the conclusions regarding the feasibility to conduct a large scale trial. What stood out for me was just how well received this feasibility study was to both patients and providers. When I saw in the limitations that the study PI was the providers boss then I think that may explain the positive views from the staff and is therefore quite a large limitation. I think this would seriously need consideration in the planning of a future trial as staff may be less receptive in other settings. Especially if it can take 30 minutes to deliver the intervention in some cases. I think the paper should include some discussion of challenges that may occur in a large scale trial that would take place across multiple settings as it does come across as overly positive

2. The statistics appear sound, but as I am not a trial statistician I think another reviewer may be better placed to comment here.

3. The author stated that the data can not be made fully available due to privacy concerns. Perhaps some further information could be provided here, is it not possible to anonymise the data?

4. The article is well written and I like how it is organised around the RE_AIM framework. A couple of points:

Page 38/68 (page 8 of manuscript) - Is there a reference for where you adapted the tools culturally using data? It would be good to include the reference here again if so

Page39/68 (page 9) - How did they train nurses in the intervention delivery? I noticed that as part of fidelity checks nurses were evaluated on how they delivered the intervention according to protocol and that the discussion reports that they were satisfied with the training - but it would be good to see in the methods how you trained them. Although brief, this is a behavioural change intervention that would require a certain standard of training that ideally should be described in the paper for replication purposes and also evaluated in order to see if the nurses are capable of delivering the intervention to protocol.

P9 - "Unusual care" heading - did you mean Usual care?!

p10 - Is home brew common in this region? If it is then this may make it difficult to estimate consumption in a larger study

p12 - Table 3 - I think there is a typo here and it should say Farmer instead of Famer

6. PLOS authors have the option to publish the peer review history of their article (what does this mean?). If published, this will include your full peer review and any attached files.

Reviewer #1: No

Reviewer #2: No

Reviewer #3: No

---

## [Author Response · Author response to Decision Letter 0]

7 Dec 2022

Reviewer Comments: 

Reviewer #1: Dear Author,

The manuscript has covered feasibility of important intervention in low resource setting. There are some major and minor comments which need to be addressed before it can be considered for the publication.

1. Major comments: Authors have mentioned that the aim of the study is to assess the feasibility however the title of study mentions it as randomized controlled trial which is inherently supposed to measure effectiveness of the intervention vis a vis control group. So authors must do statistical analysis to see if there is any significant or non significant differences in outcome measures using non-parametric tests since the sample size may not be enough for parametric tests. Otherwise it would be wrong to refer the present manuscript as RCT. The paper can be best described as feasibility study and not RCT. So please update results sections to compare outcome measures accordingly. The findings should also be discussed later.

Thank you for this feedback. This is a feasibility study and therefore are reporting the feasibility on how to conduct this trial. Our future trial will report sample size and statistical analysis for our efficacy testing analyses. 

2. Results section does not mention about findings of 15 questions in a post-

treatment survey/interview and 5 questions about satisfaction

Thank you for this feedback. Responses to these questions have been included in the results according to the topic of feasibility. 

3. The trial has higher retention rate, the reason for higher retention rate should be highlighted (e.g. home visits by investigators, financial incentives given to the project staff and number of times follow up calls were attempted). All these issues are important when intervention is used in the clinical setting.

Thank you for this feedback. Yes, we have a high retention rate in our studies due to our very proactive process in following our patients. This is a locally relevant process which we have not specifically identified because it is very site dependent. We have addressed the activities supporting our high retention rate in the discussion section.

4. The authors have not mentioned what treatment was offered for patients with alcohol use disorders, any psychiatrist consultation was sent ? referred to outpatient department for management of the problem?

Thank you for this feedback. Our treatment was a standard SBIRT with screening, brief intervention with our adapted version of the BNI called the PPKAY: Punguza Pombe Kwa Afya Yako. Only method of treatment at the time was the screening, brief intervention. 

5. The timing of intervention is ranged between 5 minutes to 30 minutes. It is important to mention average time of intervention. As this raises question about fidelity of intervention. It will be good to mention number of nursing professionals who were involved in delivery of intervention, any family members involved as some patient may be brought by law enforcement agencies while some may brought by family members. Involvement of family members are likely to impact outcome measures. Reasons-- 

Thank you for this feedback. Fidelity of this intervention was measured by reviewing audio-recorded versions of the intervention and rating it on the BI assessment checklist to ensure nurses were completing desired components of the intervention. Feedback was then provided to the nurses for future interventions. Fidelity was not measured on time given 5-30 minute intervention was dependent on patient responses, not questions asked. 

An update was made in the paper to note three research nurses performed the intervention to ensure consistency. No family members were involved at this point of the feasibility trial due to stigma. Prisoners were unable to participate. 

6. Discussion section need to be expanded to discuss the findings in reference to available literature from the region. Also clinical implications of the study if any.

Thank you for this feedback. A sentence was added to include a systematic review of brief interventions in LMIC to connect to literature from the region. No clinical implications since the study is is a feasibility pilot. The implication of this study is that the trial seems feasible and should be performed. 

Reviewer #2: 

The manuscript explores the feasibility of a pragmatic randomized adaptive controlled trial of the PPKAY intervention among subjects recruited in Tanzania. The study objectives are clearly stated; the authors proposed a block-randomized controlled trial. In the future, the effectiveness of the intervention will be assessed via a pragmatic randomized adaptive trial. My comments are as follows:

1. I still think a sample size/power statement would make the approach more rigorous. Just stating that someone included 10 participants in each arm do not make a strong case. Samples sizes for pilot/feasibility studies are requested; see below:

https://www.rds-london.nihr.ac.uk/resources/justify-sample-size-for-a-feasibility-study/

Also, the 3-stage design as proposed is not so trivial.

Thank you for this feedback. We included 10 participants in each allocated arm to ensure at least 20 participants at each stage. The study included a total of 70 participants (table 2). We used this method to replicate the average potential enrollment per week. Ensured this was addressed in sample size section. 

Reviewer #3: 

1. The data appear to support the conclusions regarding the feasibility to conduct a large scale trial. What stood out for me was just how well received this feasibility study was to both patients and providers. When I saw in the limitations that the study PI was the providers boss then I think that may explain the positive views from the staff and is therefore quite a large limitation. I think this would seriously need consideration in the planning of a future trial as staff may be less receptive in other settings. Especially if it can take 30 minutes to deliver the intervention in some cases. I think the paper should include some discussion of challenges that may occur in a large scale trial that would take place across multiple settings as it does come across as overly positive.

Thank you for this feedback. A sentence was added to the paper to discuss the potential impact of familiarity on positivity and consider including champions to address widespread dissemination challenges and maintain positivity. 

2. The statistics appear sound, but as I am not a trial statistician I think another reviewer may be better placed to comment here.

Thank you. 

3. The author stated that the data can not be made fully available due to privacy concerns. Perhaps some further information could be provided here, is it not possible to anonymize the datakjh

Thank you for this comment. The data is available with appropriate ethical consideration. Data can be obtained through regulatory Tanzanian data transfer .

4. The article is well written and I like how it is organized around the RE_AIM framework. A couple of points: Page 38/68 (page 8 of manuscript) - Is there a reference for where you adapted the tools culturally using data? It would be good to include the reference here again if so

Thank you for this comment. The paper is currently under review in another journal, but a reference was added. 

Page39/68 (page 9) - How did they train nurses in the intervention delivery? I noticed that as part of fidelity checks, nurses were evaluated on how they delivered the intervention according to protocol and that the discussion reports that they were satisfied with the training - but it would be good to see in the methods how you trained them. Although brief, this is a behavioral change intervention that would require a certain standard of training that ideally should be described in the paper for replication purposes and also evaluated in order to see if the nurses are capable of delivering the intervention to protocol.

Thank you for this feedback. Information on how nurses were trained was added. 

P9 - "Unusual care" heading - did you mean Usual care?!

Thank you for this feedback. The heading was changed to usual care. 

p10 - Is home brew common in this region? If it is then this may make it difficult to estimate consumption in a larger study.

Thank you for this comment. Homebrew is common in this region, and we have researched to determine standard drink size in the region. We can estimate a culturally relevant amount at the same and different time frames. Ensured this was addressed in the paper. 

p12 - Table 3 - I think there is a typo here and it should say Farmer instead of Famer

Thank you for this feedback. Famer was changed to Farmer.

---

## [Decision Letter · Decision Letter 1]

14 Jun 2023

PONE-D-21-39881R1Feasibility of a Pragmatic Randomized Adaptive Clinical Trial to Evaluate a Brief Negotiational Interview for Harmful and Hazardous Alcohol Use in Moshi, TanzaniaPLOS ONE

 Dear Dr. Staton,

Thank you for submitting your manuscript to PLOS ONE. We are pleased to inform you that three reviewers are satisfied that you have addressed all your concerns. However, before we can formally accept your paper for publication, there is just one outstanding requirement: Please could you include a completed copy of PLOS’ questionnaire on inclusivity in global research in your revised manuscript. Our policy for research in this area aims to improve transparency in the reporting of research performed outside of researchers’ own country or community. The policy applies to researchers who have travelled to a different country to conduct research, research with Indigenous populations or their lands, and research on cultural artefacts. Please find more information on the policy and a link to download a blank copy of the questionnaire here: https://journals.plos.org/plosone/s/best-practices-in-research-reporting. Please upload a completed version of your questionnaire as Supporting Information when you resubmit your manuscript. To be clear, we are not requesting any changes to the manuscript itself, but you may need to upload it again in order to complete the "resubmission" process.

We look forward to receiving your revised manuscript.

Kind regards,

Steve Zimmerman, PhD

Associate Editor, PLOS ONE

Journal Requirements:

Reviewers' comments:

Reviewer's Responses to Questions

**Comments to the Author**

1. If the authors have adequately addressed your comments raised in a previous round of review and you feel that this manuscript is now acceptable for publication, you may indicate that here to bypass the “Comments to the Author” section, enter your conflict of interest statement in the “Confidential to Editor” section, and submit your "Accept" recommendation.

Reviewer #1: All comments have been addressed

Reviewer #2: All comments have been addressed

Reviewer #4: All comments have been addressed

2. Is the manuscript technically sound, and do the data support the conclusions?

Reviewer #1: Yes

Reviewer #2: (No Response)

Reviewer #4: Yes

3. Has the statistical analysis been performed appropriately and rigorously? 

Reviewer #1: Yes

Reviewer #2: (No Response)

Reviewer #4: Yes

4. Have the authors made all data underlying the findings in their manuscript fully available?

Reviewer #1: Yes

Reviewer #2: (No Response)

Reviewer #4: Yes

5. Is the manuscript presented in an intelligible fashion and written in standard English?

Reviewer #1: Yes

Reviewer #2: (No Response)

Reviewer #4: Yes

6. Review Comments to the Author

Reviewer #1: The manuscript is revised as per the suggestion by reviewers, it will be interesting to see results of actual trial whenever it is conducted.

Reviewer #2: (No Response)

Reviewer #4: All comments/queries by previous reviewer have been adequately addressed. I recommend accepting this manuscript for publication

7. PLOS authors have the option to publish the peer review history of their article (what does this mean?). If published, this will include your full peer review and any attached files.

Reviewer #1: **Yes: **Roshan Bhad

Reviewer #2: No

Reviewer #4: **Yes: **Abhijit Nadkarni

---

## [Author Response · Author response to Decision Letter 1]

22 Jun 2023

We have included a completed copy your questionnaire on inclusivity in global research as Supporting Information. Though you did not request any changes to the manuscript itself, we did make the small modification to add the section suggested by the questionnaire.

We appreciate the editorial team and reviewers for their feedback.

---

## [Editor Report · Decision Letter 2]

29 Jun 2023

Feasibility of a Pragmatic Randomized Adaptive Clinical Trial to Evaluate a Brief Negotiational Interview for Harmful and Hazardous Alcohol Use in Moshi, Tanzania

PONE-D-21-39881R2

Dear Dr. Staton,

We’re pleased to inform you that your manuscript has been judged scientifically suitable for publication and will be formally accepted for publication once it meets all outstanding technical requirements.

Kind regards,

James Mockridge

Staff Editor

PLOS ONE

---

## [Editor Report · Acceptance letter]

26 Jul 2023

PONE-D-21-39881R2 

Feasibility of a Pragmatic Randomized Adaptive Clinical Trial to Evaluate a Brief Negotiational Interview for Harmful and Hazardous Alcohol Use in Moshi, Tanzania 

Dear Dr. Staton:

I'm pleased to inform you that your manuscript has been deemed suitable for publication in PLOS ONE. Congratulations! Your manuscript is now with our production department. 

Kind regards, 

on behalf of

Dr James Mockridge 

Staff Editor

PLOS ONE